# The CUL5 ubiquitin ligase complex mediates resistance to CDK9 and MCL1 inhibitors in lung cancer cells

**Shaheen Kabir[1,2,3], Justin Cidado[4], Courtney Andersen[4], Cortni Dick[4], Pei-Chun Lin[1,2], Therese Mitros[1,2], Hong Ma[1,2], Seung Hyun Baik[1,2], Matthew A Belmonte[4], Lisa Drew[4], Jacob E Corn[1,2†]\***

[1]Innovative Genomics Institute, University of California, Berkeley, Berkeley, United States; [2]Department of Molecular and Cell Biology, University of California, Berkeley, Berkeley, United States; [3]Helen Diller Family Comprehensive Cancer Center, University of California, San Francisco, San Francisco, United States; [4]Bioscience Oncology, IMED Biotech Unit, AstraZeneca, Waltham, United States

**Abstract** Overexpression of anti-apoptotic proteins MCL1 and Bcl-xL are frequently observed in many cancers. Inhibitors targeting MCL1 are in clinical development, however numerous cancer models are intrinsically resistant to this approach. To discover mechanisms underlying resistance to MCL1 inhibition, we performed multiple flow-cytometry based genome-wide CRISPR screens interrogating two drugs that directly (MCL1i) or indirectly (CDK9i) target MCL1. Remarkably, both screens identified three components (CUL5, RNF7 and UBE2F) of a cullin-RING ubiquitin ligase complex (CRL5) that resensitized cells to MCL1 inhibition. We find that levels of the BH3-only pro-apoptotic proteins Bim and Noxa are proteasomally regulated by the CRL5 complex. Accumulation of Noxa caused by depletion of CRL5 components was responsible for re-sensitization to CDK9 inhibitor, but not MCL1 inhibitor. Discovery of a novel role of CRL5 in apoptosis and resistance to multiple types of anticancer agents suggests the potential to improve combination treatments.
DOI: https://doi.org/10.7554/eLife.44288.001

**\*For correspondence:**
jacob.corn@biol.ethz.ch

**Present address:** †Department of Biology, ETH Zürich, Zürich, Switzerland

## Introduction

Cancer cells frequently manipulate the intrinsic apoptotic pathway to evade cell death and expand their proliferative capacity. Aberrant increases in levels of anti-apoptotic proteins in the BCL-2 family, such as amplification of *MCL1*, have been widely implicated in the transformation of cancer cells and the development of resistance to current therapies (*Kelly and Strasser, 2011*). BCL-2 family members are classified based on the conservation of their BCL-2 homology (BH) domains: multi-domain proteins BAK, BAX and BOK serve as apoptosis executors in the mitochondria; proteins containing only the BH3 domain (BH3-only) promote BAK/BAX activation. Anti-apoptotic proteins such as MCL1, Bcl-xL and BCL-2 inhibit apoptosis by antagonistic binding to pro-apoptotic BH3-only proteins as well as BAK and BAX (*Czabotar et al., 2014*).

High-resolution investigation of somatic copy number alterations has revealed that gene amplification of *MCL1* and *BCL2L1* (Bcl-xL) are key determinants of survival in many cancers, including breast cancer, non-small cell lung cancer (NSCLC), multiple myeloma, acute myeloid leukemia, and B-cell acute lymphoblastic leukemia (*Goodwin et al., 2015*; *Koss et al., 2013*; *Xiao et al., 2015*; *Zhang et al., 2011*). Amplification of *MCL1* is a prognostic indicator for disease severity and progression, making it an attractive therapeutic target (*Campbell et al., 2018*; *Yin et al., 2016*).

In an effort to restrict the action of anti-apoptotic proteins, numerous compounds have been developed that mimic BH3-only proteins (BH3-mimetics). Unfortunately, the first BH3-mimetics that

**eLife digest** Organisms keep their tissues healthy by instructing damaged or unwanted cells to kill themselves via a controlled process known as apoptosis. Cancer cells, however, are able to evade death by increasing the level of proteins that block apoptosis, such as MCL1.

Researchers have recently developed new drugs that can inhibit the action of the MCL1 protein. But a number of cancers have become resistant to these inhibitors. So, one important question is whether other proteins in cancer cells could be drugged, together with MCL1, to overcome or even avoid this resistance.

Now, Kabir et al. have addressed this question by searching the genome of human lung cancer cells, which were resistant to treatment, for targets that could improve the performance of two MCL1 inhibitors. This involved reducing the level of every protein in these cells one by one using a genetic technique known as CRISPR-Cas9, and looking for cells that lost their resistance to the MCL1 inhibitor.

From these genetic screens, Kabir et al. identified three proteins that are part of complex called CRL5. Inactivating this protein complex caused cancer cells to become more sensitive to the MCL1 inhibitor. Further biochemical experiments showed that CRL5 may contribute to drug resistance by reducing the levels of two proteins that promote apoptosis.

These findings suggest that inhibiting CRL5 in combination with MCL1 could combat drug resistance. Although there are currently no drugs against CRL5, future experiments determining how CRL5 and MCL1 are linked could identify new drug targets and improve existing cancer treatments.
DOI: https://doi.org/10.7554/eLife.44288.002

specifically antagonized Bcl-xL were associated with significant thrombocytopenia, thus complicating their therapeutic use (*Lessene et al., 2013*; *Leverson et al., 2015a*; *Tao et al., 2014*). Small-molecule inhibition of MCL1 has recently gained significant attention (*Figure 1A*), and compounds that selectively target MCL1 are currently in clinical trials (*Abulwerdi et al., 2014*; *Burke et al., 2015*; *Caenepeel et al., 2018*; *Kotschy et al., 2016*; *Leverson et al., 2015b*; *Tron et al., 2018*; Phase I Study of S64315 Administred Intravenously in Patients With Acute Myeloid Leukaemia or Myelodysplastic Syndrome). Promising reports of direct BH3-mimetic MCL1 inhibitors in preclinical hematological malignancies show potent efficacy with low cytotoxicity (*Kotschy et al., 2016*; *Leverson et al., 2015b*). However, assessment of MCL1 inhibitors in solid breast tumors showed little single agent activity unless combined with a chemotherapeutic agent (*Merino et al., 2017*). Co-dosing Bcl-xL and MCL1 inhibitors to achieve effective treatment may be complicated by severe accompanying side effects.

Beyond direct inhibitors of the BCL2 family of proteins, inhibitors of cyclin-dependent kinase 9 (CDK9) can indirectly target MCL1. CDK9 inhibition restricts transcription elongation thus exploiting all mRNAs and proteins that have short-lived half-lives. Due to its short half-life, MCL1 is one of several targets that is particularly susceptible to acute CDK9i treatment, and other (proto-)oncogenes such as MYC are also CDK9i targets (*Figure 1A*) (*Akgul et al., 2000*; *Gregory et al., 2015*; *Huang et al., 2014a*; *Lemke et al., 2014*). Although CDK9 inhibition suppresses MCL1 expression, it does not affect levels of other anti-apoptotic proteins such as Bcl-xL, exposing a potential vulnerability in CDK9i treatment such that cancers may already have or develop a mode of resistance. Selective CDK9 inhibitors have shown promising results in preclinical murine models; however, they have a limited therapeutic window and must be acutely dosed for clinical applications due to their global effects on transcription (*Garcia-Cuellar et al., 2014*; *Hellvard et al., 2016*). In order to truly harness the power of CDK9 inhibitors (CDK9i) or MCL1 inhibitors (MCL1i), it is imperative to uncover additional targets that may sensitize cells to these treatments.

Since CDK9i shares at least one of its targets with MCL1i, we reasoned that a genome wide search could uncover shared factors that modulate the therapeutic activity of these compounds and suggest approaches to resensitize otherwise resistant tumor cells. As lung cancer is the leading cause of cancer mortality and most NSCLC patients develop resistance to first-line treatment, we performed genome-wide CRISPR inhibition (CRISPRi) screens in a NSCLC line resistant to both CDK9 and MCL1 inhibition (*Siegel et al., 2016*). We discovered that disruption of multiple members

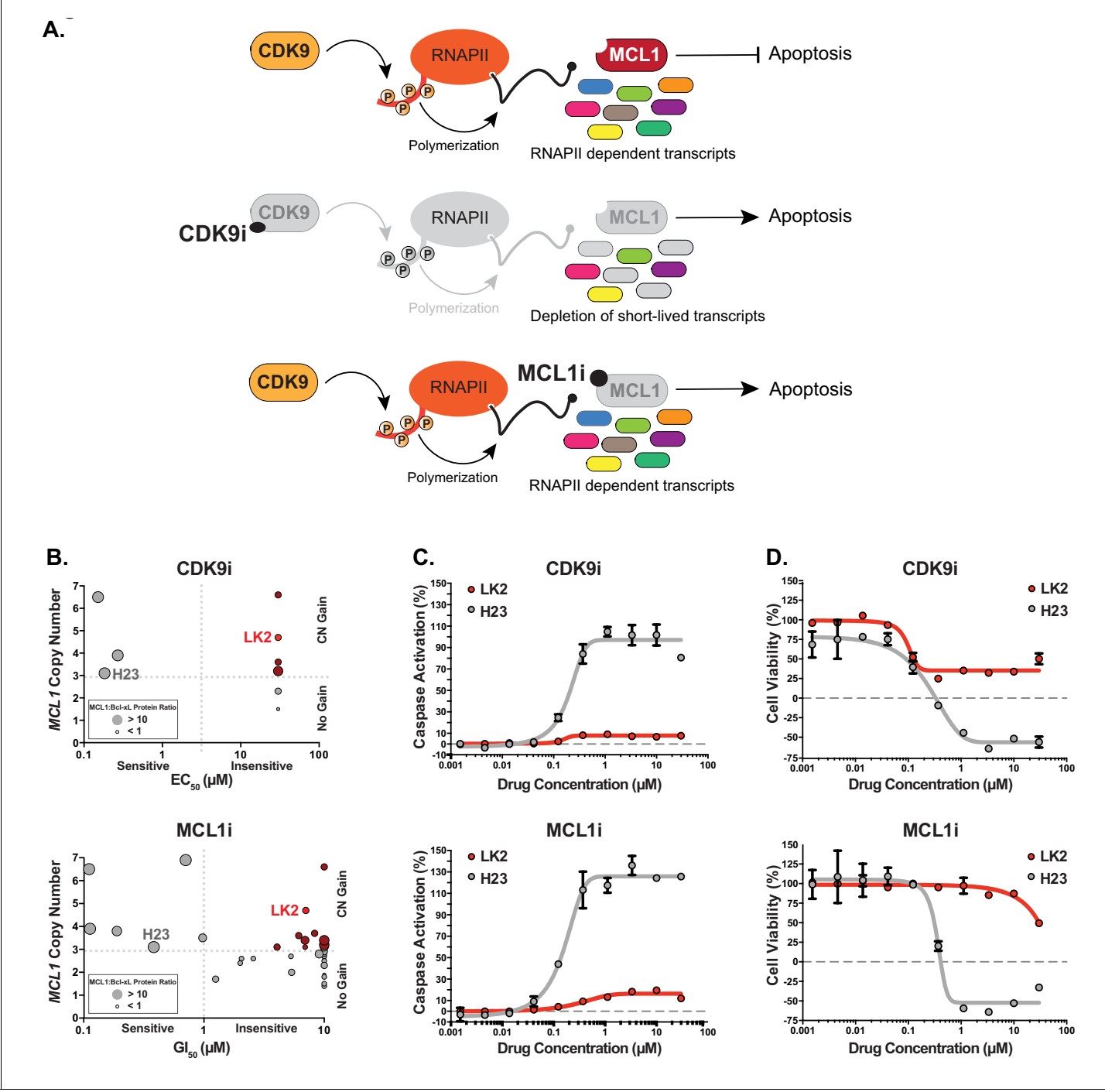

**Figure 1.** Several *MCL1*-amplified NSCLC lines are resistant to treatment with CDK9i or MCL1i. (**A**) Schematic illustrating the mechanism of action of CDK9 and MCL1 inhibitors. The CDK9 inhibitor (CDK9i) inhibits transcription elongation, thus mRNAs with short half-lives such as MCL1 are highly susceptible to acute CDK9 inhibition. The MCL1 inhibitor (MCL1i) is a BH3-mimetic that binds directly to MCL1. (**B**) Graphical representation of a panel of cell lines depicting their *MCL1* copy number, their ratio of MCL1:Bcl-xL protein and whether they are sensitive to the drug treatment indicated. $EC_{50}$ values plotted for a 6 hr CDK9i treatment (top graph) derived from Caspase-Glo 3/7 assays. $GI_{50}$ values plotted for a 24 hr MCL1i treatment (bottom graph) using CellTiter-Glo. Maroon circles indicate cell lines resistant to drug despite being MCL1-amplified. Highlighted in bright red is a resistant cell line (LK2) used for further study in this report and a sensitive cell line (H23) is shown in gray. (**C**) Dose response curves of LK2 and H23 treated with CDK9i (top) and MCL1i (bottom). Caspase activation was measured at 6 hr post drug treatment at the indicated concentrations by CaspaseGlo 3/7 and normalized to a positive control containing inhibitors of MCL1, BCL2 and Bcl-xL. (**D**) Cell viability curves of the resistant LK2 and sensitive H23 lines 24

*Figure 1 continued on next page*

*Figure 1 continued*

hr following drug treatment with CDK9i (top) or MCL1 (bottom) at increasing concentrations as indicated. Viability was measured using the Cell Titer Glo assay normalized to a DMSO control.

DOI: https://doi.org/10.7554/eLife.44288.003

of the cullin 5-RING ubiquitin ligase (CRL5) complex markedly resensitized cells to both CDK9i and MCL1i. The CRL5 complex targets pro-apoptotic BH3-only proteins Bim and Noxa for proteasomal degradation. Epistatic knockdown of Bim or Noxa in a CUL5 knockout background rendered cells once again resistant to CDK9i, but did not affect increased sensitivity to MCL1i. Our data indicate that members of the CRL5 complex modulate the apoptotic threshold and could be attractive targets for future combination therapy to treat otherwise resistant NSCLC.

## Results

### *MCL1*-amplified NSCLC lines resistant to CDK9i and MCL1i have increased Bcl-xL

We assessed a panel of NSCLC lines for their sensitivity to CDK9i (AZD5576) or MCL1i (AZD5991) (*Figure 1B* and *Table 1*). Several cell lines with amplified *MCL1* (copy number ≥3) were highly sensitive to CDK9i and MCL1i, consistent with overexpression of MCL1 to escape apoptosis. We also found several NSCLC lines resistant to CDK9i and MCL1i despite being *MCL1*-amplified. Increased Bcl-xL expression correlated with the *MCL1*-amplified lines that were resistant to CDK9i and MCL1i. Conversely, *MCL1*-amplified cell lines with low levels of Bcl-xL were sensitive to CDK9i and MCL1i.

Both MCL1i and CDK9i induce apoptosis and cell death in a sensitive cell line (H23) after just 6 hr, with saturated cell death and caspase activation at 1 μM. Conversely, a resistant line (LK2) was about 100-fold more resistant to CDK9i and MCL1i (*Figure 1C*). After extended treatment (24 hr), CDK9i induced a significant decrease in cell viability in both the resistant and sensitive cell lines, highlighting off-target toxicity presumably stemming from global transcriptional arrest independent of targeting short half-life transcripts such as MCL1 and MYC (*Figure 1D*). This emphasizes the need for a tightly regulated treatment window for CDK9i to avoid non-specific cell death.

### Genome-wide CRISPRi screens identify factors that synergize with CDK9 and MCL1 inhibition

While Bcl-xL amplification correlates with resistance to MCL1-targeting drugs, preclinical studies suggest dual inhibition of MCL1 and Bcl-xL may not be a clinically viable option due to toxic side

**Table 1.** A subset of resistant and sensitive NSCLC lines.

For each cell line, the table indicates *MCL1* copy number, MCL1:Bcl-xL protein ratio and $EC_{50}$ concentrations for both CDK9i and MCL1i treatments. Resistant cell lines are in dark pink shaded rows; sensitive cell lines are in light pink shaded rows.

| | MCL1 copy number | Mcl1:BclxL protein ratio | MCL1i $GI_{50}$ (μM) | MCL1i caspase $EC_{50}$ (μM) | CDK9i caspase $EC_{50}$ (μM) |
|---|---|---|---|---|---|
| SKLU1 | 1.5 | 0.41 | 10.000 | 30.000 | 30.000 |
| HCC827 | 2.3 | 4.38 | 10.000 | 30.000 | 30.000 |
| H460 | 3.2 | 16.58 | 10.000 | 30.000 | 30.000 |
| H1734 | 3.6 | 2.63 | 6.150 | 20.056 | 30.000 |
| LK2 | 4.7 | 3.71 | 7.040 | 30.000 | 30.000 |
| H1395 | 6.6 | 3.79 | 10.000 | 15.404 | 30.000 |
| H23 | 3.1 | 39.59 | 0.383 | 0.198 | 0.183 |
| H2110 | 3.9 | 18.29 | 0.113 | 0.226 | 0.270 |
| H1568 | 6.5 | 14.26 | 0.111 | 0.3 | 0.151 |

DOI: https://doi.org/10.7554/eLife.44288.004

effects. We sought to identify alternate pathways that could resensitize resistant LK2 lung cancer cells to CDK9i or MCL1i. Prolonged CDK9i treatment induces non-specific cell death, possibly due to its polypharmacology, and so a growth-based screen measuring cell abundance after extended compound exposure was inappropriate. Instead, we developed a positive selection FACS-based screen for apoptosis during acute exposure to maximize on-target cell death and minimize non-specific cell death (*Figure 2A*).

We exposed a clonal LK2 CRISPRi cell line transduced with a genome-wide CRISPRi-v2 guide RNA library (Materials and methods and *Figure 2A*) to either 3 μM CDK9i for 6 hr or 1 μM MCL1i for 12 hr. Per-cell apoptosis was assayed with 0.5 μM Cell Event, which fluoresces upon cellular caspase activation, and apoptosing cells were separated by genome-scale fluorescence activated cell sorting (FACS). To identify vehicle effects, we also performed duplicate genome-wide sorts with DMSO. Bulk untreated cells were harvested on the same day as the FACS sort to provide an accurate 'background' sampling of sgRNAs and eliminate confounding effects from essential genes that had dropped out of the population. We measured quantitative differences in sgRNA frequency by deep sequencing and integrated enriched or depleted sgRNAs into gene-level hits by comparing sorted samples to the untreated control using ScreenProcessing (*Figure 2—source datas 1–3*) and MAGeCK (*Figure 2—source datas 4–7*) (*Horlbeck et al., 2016*; *Li et al., 2014*).

No significant gene-level hits were detected in the DMSO control (*Figure 2—figure supplement 1B*), indicating that gene calls in CDK9i- or MCL1i-treated samples were derived from drug action. We found multiple sgRNAs targeting Bcl-xL led to re-sensitization of LK2 cells to CDK9i and MCL1i (*Figure 2B–C*), confirming reports that depletion of both MCL1 and Bcl-xL initiates apoptosis (*Goodwin et al., 2015*; *Xiao et al., 2015*; *Zhang et al., 2011*). Downregulation of the mitochondrial porin VDAC2 also promoted apoptosis in drug-treated cells (*Figure 2B–C*), consistent with its proposed role in BAK/BAX sequestration at the mitochondrial membrane (*Cheng et al., 2003*; *Chin et al., 2018*; *Lauterwasser et al., 2016*). *EIF4G2* (DAP5) was identified as a hit in the CDK9i screen (*Figure 2B*), and this gene has been implicated in stimulating cap-independent translation of anti-apoptotic protein BCL-2 (*Marash et al., 2008*).

Comparing the most significantly enriched genes from both screens showed an overlap between MCL1i and CDK9i. The CDK9i screen had many more hits, consistent with its polypharmacology of transcriptional inhibition (*Figure 2—figure supplement 1C*). Despite separate mechanisms of drug action and potentially distinct targets, significantly more of the top resensitization hits were shared between CDK9i and MCL1i than are expected by chance (exact hypergeometric test $p<6.4\times10^{-9}$). These shared hits are part of two physical complexes, one potentially involved in specialized translation and the other involved in protein degradation. First, knockdown of multiple components of the eukaryotic translation initiation factor 3 (eIF-3) complex (EIF3H and EIF3M) resensitize LK2 cells to CDK9i and MCL1i (*Figure 2B–C* and *Figure 2—figure supplement 1D–E*). eIF-3 is reported to bind a highly specific program of messenger RNAs involved in cell proliferation and apoptosis (*Lee et al., 2015*). We speculate that this complex could be involved in mediating translational activation of certain anti-apoptotic proteins or repression of pro-apoptotic proteins. Regulation of apoptosis by eIF-3 warrants further investigation, but we opted to focus on the second, better-studied complex.

Knockdown of multiple members of a cullin-RING ubiquitin ligase complex (CRL) including CUL5, UBE2F and RNF7 resensitize LK2 cells to CDK9i and MCL1i (*Figure 2B–C* and *Figure 2—figure supplement 1D–E*). The cullin 5 (*CUL5*) scaffold was identified as a top resensitizing hit in both screens. Repression of *UBE2F* was a sensitizing hit in the MCL1i screen (*Figure 2C* and *Figure 2—figure supplement 1E*). Knockdown of Ring Finger Protein 7 (*RNF7*), a catalytic subunit that binds to the CUL5 scaffold, also sensitizes cells to both CDK9i and MCL1i (*Figure 2—figure supplement 1D–E*).

## Depletion of the CUL5-RNF7-UBE2F ubiquitin ligase complex resensitizes LK2 cells to treatment with CDK9i or MCL1i

Cullin-RING ligases are emerging as attractive cancer targets and a novel class of small molecule neddylation inhibitors have recently been developed (*Soucy et al., 2009*). While the cullin 3 complex (CRL3) is more typically associated with cancer phenotypes, there is little data on the function of CRL5 in tumorigenesis or resistance. We therefore sought to further examine the role of the CRL5 complex in resensitizing LK2 cells to CDK9 and MCL1 inhibition.

CUL5 serves as a protein scaffold in the CRL5 complex, forming a platform for RNF7, UBE2F, Elongin B/C, and a substrate adaptor to recruit and ubiquitinate a target substrate (*Figure 3A*)

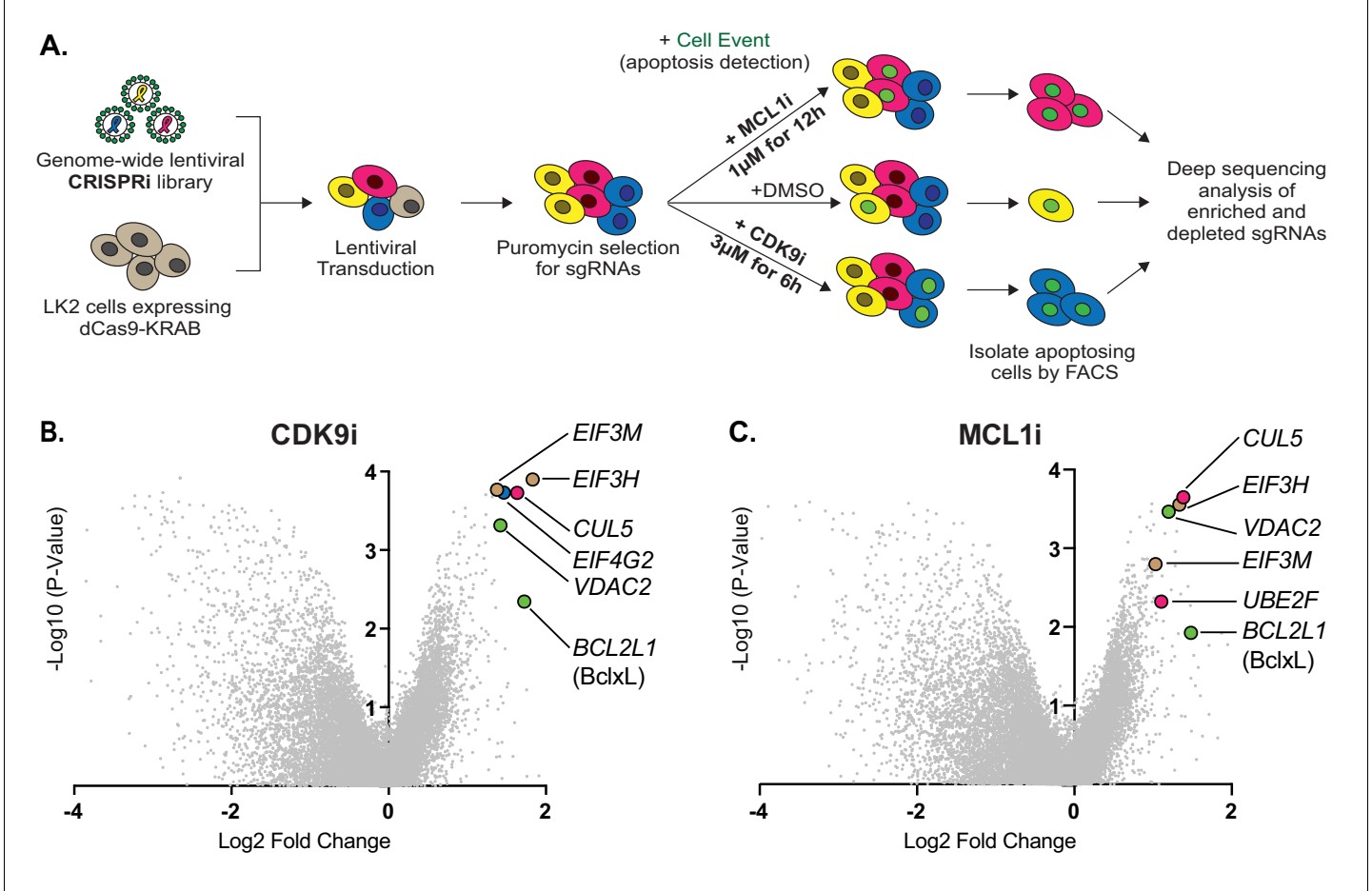

**Figure 2.** Genome-wide CRISPRi screens identify factors that resensitize lung cancer cells to inhibition of CDK9 or MCL1. (**A**) Schematic outlining the genome-wide CRISPRi screen in LK2 cells. Cells were exposed to acute drug treatments, fixed and FACS-sorted using the fluorogenic apoptotic detection reagent Cell Event. Enriched and depleted sgRNAs were identified by next-generation sequencing. (**B + C**) Volcano plots showing sgRNA-targeted genes significantly enriched or depleted in the apoptosing cell population following treatment with CDK9i (**B**) or MCL1i (**C**). Average of two independent experiments is graphed. Green highlighted points indicate genes with a known role in apoptosis that had a significant fold change over background. Magenta points highlight members of the CUL5-RNF7-UBE2F ubiquitin complex that were significantly enriched. Beige highlighted points are members of the *Eukaryotic Translation Initiation Factor 3* (eIF3) complex. Blue point on CDK9i volcano plot highlights *EIF4G2*, a gene that may be involved in cap-independent translation of Bcl-xL.

DOI: https://doi.org/10.7554/eLife.44288.005

The following source data and figure supplement are available for figure 2:

**Source data 1.** ScreenProcessing: sgRNA counts.
DOI: https://doi.org/10.7554/eLife.44288.007
**Source data 2.** ScreenProcessing: sgRNA phenotype scores.
DOI: https://doi.org/10.7554/eLife.44288.008
**Source data 3.** ScreenProcessing: gene phenotype scores.
DOI: https://doi.org/10.7554/eLife.44288.009
**Source data 4.** MAGeCK: sgRNA summary for CDK9i.
DOI: https://doi.org/10.7554/eLife.44288.010
**Source data 5.** MAGeCK: gene summary for CDK9i.
DOI: https://doi.org/10.7554/eLife.44288.011
**Source data 6.** MAGeCK: sgRNA summary for MCL1i.
DOI: https://doi.org/10.7554/eLife.44288.012
**Source data 7.** MAGeCK: gene summary for MCL1i.
DOI: https://doi.org/10.7554/eLife.44288.013
**Figure supplement 1.** Performing the genome-wide CRISPRi screens and analysis of results.
DOI: https://doi.org/10.7554/eLife.44288.006

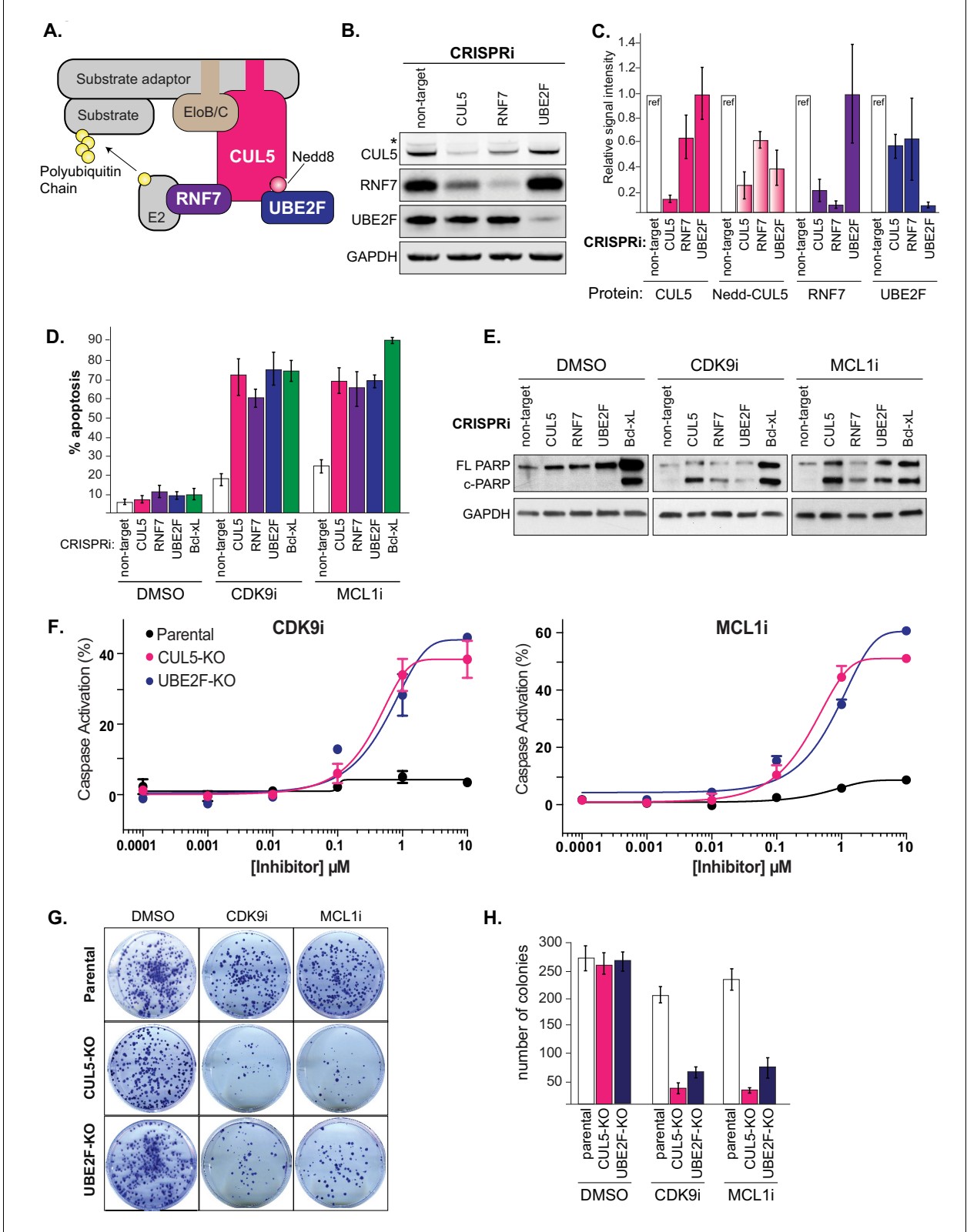

**Figure 3.** Depletion of the CUL5-RNF7-UBE2F ubiquitin complex induces apoptosis upon treatment with CDK9i or MCL1i. (**A**) Schematic depicting the CUL5 ubiquitin ligase scaffold and its interacting partners. (**B**) Western blots confirming effective knockdown of CUL5, RNF7 and UBE2F by stable lentiviral expression of dCas9-KRAB and corresponding sgRNAs. Asterisk indicates neddylated CUL5. GAPDH serves as loading control. (**C**) Quantification of western blots as in B. Error bars show standard deviations from three independent biological replicates. (**D**) Induction of apoptosis

*Figure 3 continued on next page*

Figure 3 continued

when cells depleted of CUL5, RNF7 or UBE2F as in (B) are treated with CDK9i (3 µM for 6 hr), MCL1i (1 µM for 12 hr) or DMSO. Knockdown of Bcl-xL serves as a positive control for induction of apoptosis. Percentage of apoptosis determined by flow cytometry detection of the fluorogenic apoptotic detection reagent, Cell Event. Error bars are standard deviations of three independent biological replicates. (E) Western blotting for cleaved PARP (c-PARP) serves as orthogonal readout for induction of apoptosis following knockdown of target genes and treatment with 3 µM CDK9i for 6 hr or 1 µM MCL1i for 12 hr. Knockdown of Bcl-xL is included as a positive control, however due to extreme toxicity of the combination treatment, cells were harvested 3 hr after treatment with 3 µM CDK9i or 0.5 hr after treatment with 1 µM MCL1i. GAPDH, loading control. (F) Dose response curves of caspase induction showing resensitization of *CUL5* knockout (CUL5-KO c3) and *UBE2F* knockout (UBE2F-KO c1) lines as compared to parental LK2 cells when treated with CDK9i (top) and MCL1i (bottom). Caspase activation was measured at 10 hr post drug treatment at the indicated concentrations by CaspaseGlo 3/7 and normalized to a positive control containing inhibitors of MCL1, BCL2 and Bcl-xL. (G) Colony forming potential is decreased in *CUL5*-KO and *UBE2F*-KO as compared to parental LK2 cells after 1 µM treatment with CDK9i or MC1i for 8 hr. Representative images shown of colonies stained 2 weeks after drug treatment with 0.5% crystal violet in 6% glutaraldehyde. Experiments performed in triplicate. (H) Quantification of colonies as in (G), error bars represent standard deviations of three independent experiments.

DOI: https://doi.org/10.7554/eLife.44288.014

The following source data and figure supplements are available for figure 3:

**Source data 1.** Analysis of apoptosis following knockdown of putative CUL5 substrate adaptors challenged with CDK9i or MCL1i.
DOI: https://doi.org/10.7554/eLife.44288.017
**Figure supplement 1.** Generation of *CUL5* and *UBE2F* knockouts also resensitize cells to CDK9 and MCL1 inhibition.
DOI: https://doi.org/10.7554/eLife.44288.015
**Figure supplement 2.** Knockdown of Elongin B but not Elongin C also resensitizes cells to CDK9 and MCL1 inhibition.
DOI: https://doi.org/10.7554/eLife.44288.016

(*Kamura et al., 2004*) (*Huang et al., 2009*) (*Mahrour et al., 2008*; *Okumura et al., 2016*). We made stable dCas9-KRAB LK2 cell lines that expressed an individual CRISPRi sgRNA to knockdown CUL5, RNF7, UBE2F, or a non-targeting (NT) control (*Figure 3B*). CUL5 and RNF7 appeared to be dependent on each other for stability such that when either protein was knocked down, levels of the corresponding protein were also diminished (*Figure 3C*). Depletion of UBE2F resulted in the disappearance of a higher molecular weight band corresponding to neddylated CUL5 (*Figure 3B–C*).

Using two independent stable LK2 CRISPRi lines targeting CUL5, RNF7, UBE2F or Bcl-xL, we validated that individual knockdowns of each gene resensitized the cell to both MCL1i and CDK9i (*Figure 3D* and *Figure 3—figure supplement 1A*). Bcl-xL served as a positive control based on previous reports that dual inhibition of Bcl-xL and MCL1 induces apoptosis (*Goodwin et al., 2015*; *Xiao et al., 2015*; *Zhang et al., 2011*). Assessing cleaved PARP (c-PARP) levels, we found that depletion of CUL5, RNF7, UBE2F or Bcl-xL significantly induced PARP cleavage when combined with CDK9i or MCL1i (*Figure 3E*). In the absence of drug, knockdown of Bcl-xL alone led to substantial apoptosis but this was not the case for knockdown of CUL5, RNF7 or UBE2F (*Figure 3E*). This suggests that inactivation of CRL5 could be less toxic than inhibition of Bcl-xL and that inhibition of CRL5 may be better suited to co-administration with CDK9 and MCL1 inhibitors.

To further validate the synergy between inhibition of CRL5 and CDK9i or MCL1i, we made isogenic CRISPR-Cas9 knockouts of *CUL5* and *UBE2F* in the LK2 background (Materials and methods and *Figure 3—figure supplement 1B–C*). Cells did not tolerate extended culturing of CRISPRi-mediated stable knockdowns of *RNF7*, but knockouts of *CUL5* and *UBE2F* were viable. Challenging *CUL5* or *UBE2F* knockout cells with CDK9i or MCL1i induced high levels of apoptosis at 1 µM of either compound, whereas wild type cells showed no response even at 10 µM (*Figure 3F*, *Figure 3—figure supplement 1D*). To assess effects on long-term survival and proliferation, we performed clonogenic assays where the *CUL5* and *UBE2F* knockout clones were exposed to an acute treatment of CDK9i or MCL1i (*Figure 3G–H*). Drug treatment greatly reduced colony-forming ability in the *CUL5* and *UBE2F* knockout cells, indicating efficient and persistent resensitization of these cells. Taken together, multiple lines of evidence indicate that the resistance of LK2 cells to both CDK9 and MCL1 inhibition can be overcome by depletion of the CRL5 complex.

To further investigate how CRL5 protects LK2 cells from CDK9i and MCL1i, we asked whether depletion of Elongin B and Elongin C also synergized with these drugs. Individual knockdown of Elongin B by CRISPRi-induced apoptosis at levels similar to that of CUL5 knockdown, but knockdown of Elongin C had minimal effect (*Figure 3—figure supplement 2A–B*). Elongin B and C have often

been described as functioning in a complex, but our results suggest a potential separation of function (*Okumura et al., 2012*).

In an attempt to find the CRL5 substrate adaptor responsible for resensitization to MCL1 and CDK9 inhibitors, we used CRISPRi to individually knock down a large number of CUL5 substrate adaptors annotated in literature and tested them for synergy with MCL1i and CDK9i (*Okumura et al., 2016*). None of the literature-proposed substrate adaptors analyzed induced apoptosis following CDK9i or MCL1i treatment (*Figure 3—figure supplement 2C* and *Figure 3—source data 1*), but we cannot rule out redundancy between substrate adaptors. The CRL5 substrate adaptors responsible for resensitization to MCL1 inhibition remain to be identified.

## The CRL5 complex regulates levels of the BH3-only apoptotic sensitizers Bim and Noxa

We searched for the substrate of CRL5 that potentiates resensitization to CDK9 or MCL1 inhibition. p53, a master regulator of apoptosis, can be a target of CRL5 during viral infection (*Cai et al., 2006*; *Querido et al., 2001*; *Sato et al., 2009*). However, loss of CUL5 did not lead to accumulation of p53 (*Figure 3—figure supplement 2D*), nor affected MCL1 and Bcl-xL levels (*Figure 3—figure supplement 2E*).

Pro-apoptotic proteins are required for efficient induction of cell death, and CRL5 could target pro-apoptotic proteins for proteasomal degradation. Indeed, Noxa was recently proposed to be a substrate of CRL5 (*Zhou et al., 2017*). We interrogated the levels of all eight BH3-only proteins using individual CRISPRi knockdowns of *CUL5*, *RNF7*, *UBE2F* and a NT control. Knockdown of CRL5 components increased protein levels of two BH3-only proteins, Noxa and Bim (*Figure 4A*). After treatment with CDK9i or MCL1i, we also found that *CUL5* knockdown had increased levels of both proapoptotic proteins Noxa and Bim while all other BH3-only proteins remained unchanged (*Figure 4B–C*). Interestingly, CDK9i treatment alone reduced levels of Noxa and Bim through transcriptional downregulation, presumably related to the half-lives of these mRNAs (*Figure 4D*). Knockdown of CUL5 together with CDK9i rescued protein levels of Noxa and Bim but not mRNA levels, consistent CRL5-mediated post-translational degradation of Noxa and Bim.

We used epoxomicin (proteasomal inhibitor) and folimycin (lysosomal inhibitor) to confirm that CRL5 targets Noxa and Bim for proteasomal degradation. Folimycin treatment did not affect the abundance of Noxa or Bim (*Figure 4E*). However, epoxomicin treatment causes accumulation of Bim in LK2 parental cells. Epoxomicin treatment had no additional effect on Bim in *CUL5* knockout cells. Hence, under these conditions CRL5 appears to be the primary ligase for Bim. Noxa accumulates in both parental and *CUL5* knockout cells after epoxomicin treatment (*Figure 4E*). These data indicate that Noxa is proteasomally regulated, but suggest that Noxa can be targeted by additional ubiquitin ligases. We were unable to demonstrate direct ubiquitination of Noxa and Bim by CRL5 via ubiquitin co-immunoprecipitation (*Figure 4—figure supplement 1*), leaving open the possibility that these proteins are indirect substrates.

## Noxa is required to resensitize CUL5-deficient cells to CDK9 inhibition

We examined whether Noxa and/or Bim were required to resensitize CUL5-CRISPRi cells to CDK9i or MCL1i using RNAi. If induction of apoptosis in CUL5-deficient cells requires Noxa and/or Bim, then removal of either or both factors should make the cells re-resistant to CDK9i and MCL1i. Knocking down BAK, the apoptosis effector required for cell death, prevented apoptosis induced by CDK9i or MCL1i in CUL5-CRISPRi cells (*Figure 5A–B* and *Figure 5—figure supplement 1*).

Knocking down Noxa in CUL5-CRISPRi cells almost completely prevented CDK9i-induced apoptosis (*Figure 5A* and *Figure 5—figure supplement 1A*). Knocking down Bim in CUL5-CRISPRi cells did not prevent CDK9i-induced apoptosis, and simultaneously knocking down both Noxa and Bim in CUL5-CRISPRi cells did not further inhibit apoptosis beyond what was observed with knockdown of Noxa alone. These data indicate that in CDK9i-treated cells lacking CUL5, initiation of apoptosis is primarily dependent on Noxa. Surprisingly, knocking down Noxa and/or Bim in CUL5-CRISPRi cells exposed to MCL1i still led to high levels of apoptosis (*Figure 5B* and *Figure 5—figure supplement 1B*). The difference in apoptotic dependence between MCL1i and CDK9i could be due to transcriptional downregulation of Noxa and Bim during CDK9i (*Figure 4D*) or from downregulation of other pro-apoptotic factors, whose loss in conjunction with Noxa depletion leads to abrogation of

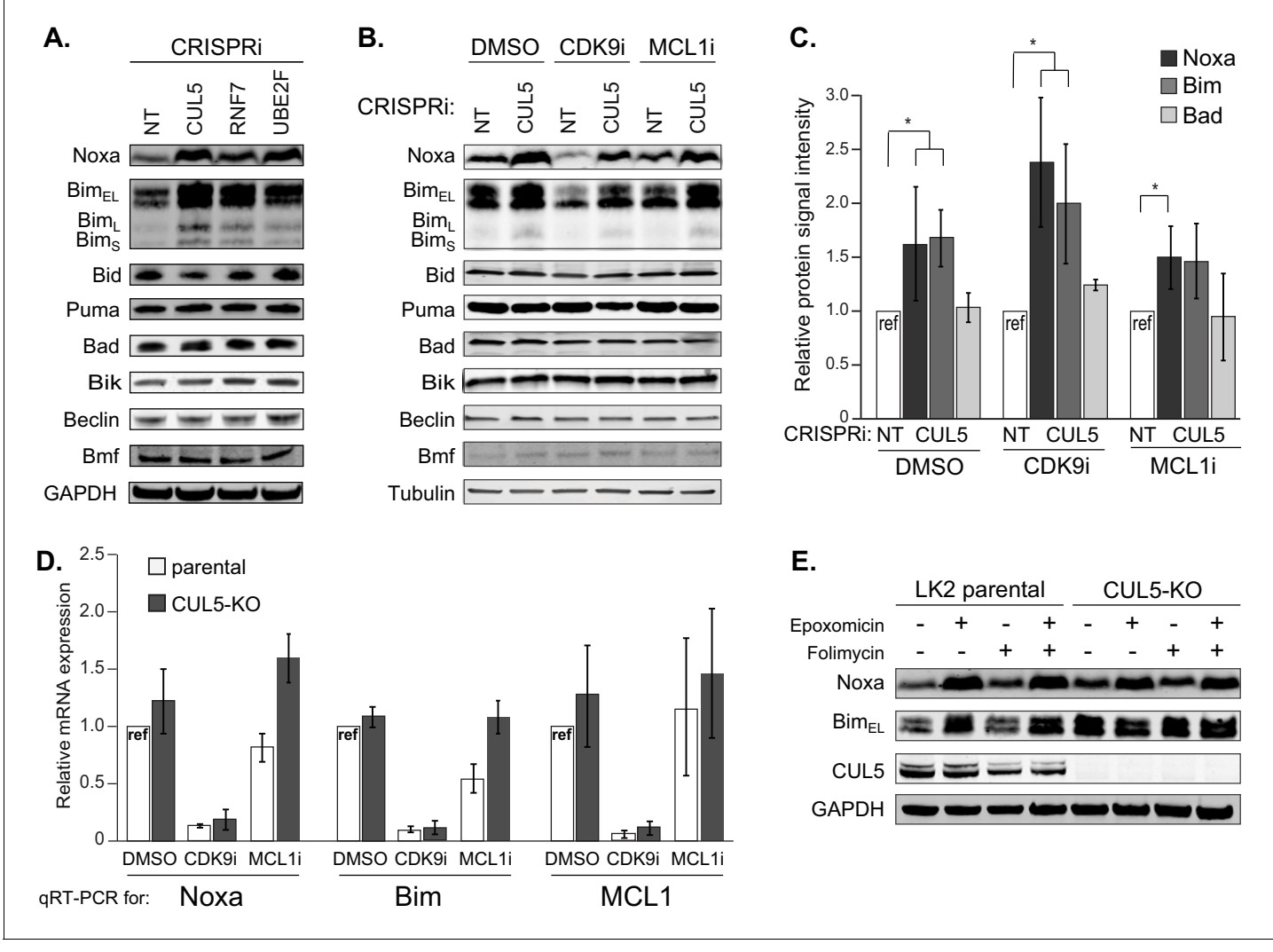

**Figure 4.** The CUL5-RNF7-UBE2F ubiquitin complex regulates levels of BH3-only apoptotic sensitizers Bim and Noxa. (A) Western blotting of all BH3-only proteins in cell lines with depleted CUL5, RNF7 or UBE2F show increased levels of Bim and Noxa as compared to cells transduced with a non-targeting (NT) sgRNA. GAPDH serves as the loading control. Blots are representative of three independent biological replicates. (B) Western blot of all BH3-only proteins in NT and CUL5-depleted cells treated with DMSO, CDK9i (3 µM for 6 hr) or MCL1i (1 µM for 12 hr). Tubulin serves as loading control. Blots are representative of three independent biological replicates. (C) Quantification of western blots as in (B). Error bars derived from standard deviations of three biological replicates consisting of independently treated and isolated protein samples. Asterisk indicates p value < 0.05 as determined by a paired student T-Test. (D) qRT-PCR showing relative mRNA levels of Noxa, Bim and MCL1 in LK2 parental and CUL5-KO c1 cells treated with DMSO, CDK9i (3 µM for 6 hr) or MCL1i (3 µM for 6 hr). Error bars show standard deviations from three biological replicates on independently treated and isolated RNA samples. (E) Western blot of LK2 and CUL5-KO c1 cells treated for 6 hr with epoxomicin (100 µM), or folimycin (100 µM) or both epoxomicin and folimycin (100 µM each). GAPDH, loading control. Blots are representative of three independent biological replicates.
DOI: https://doi.org/10.7554/eLife.44288.018

The following figure supplement is available for figure 4:

**Figure supplement 1.** Indirect regulation of Bim and Noxa by the CUL5-RNF7-UBE2F complex.
DOI: https://doi.org/10.7554/eLife.44288.019

apoptosis. The mechanism underlying CRL5-mediated resensitization to MCL1i is still unclear, and our data highlight differences between direct inhibition of MCL1 and indirect downregulation via CDK9 that can also affect other factors.

In conclusion, we find that depletion of key components of the CRL5 family of ubiquitin ligases sensitizes otherwise resistant cells to CDK9 and MCL1 inhibition. We show that the CRL5 complex targets pro-apoptotic proteins Noxa and Bim for degradation, and increased Noxa abundance upon

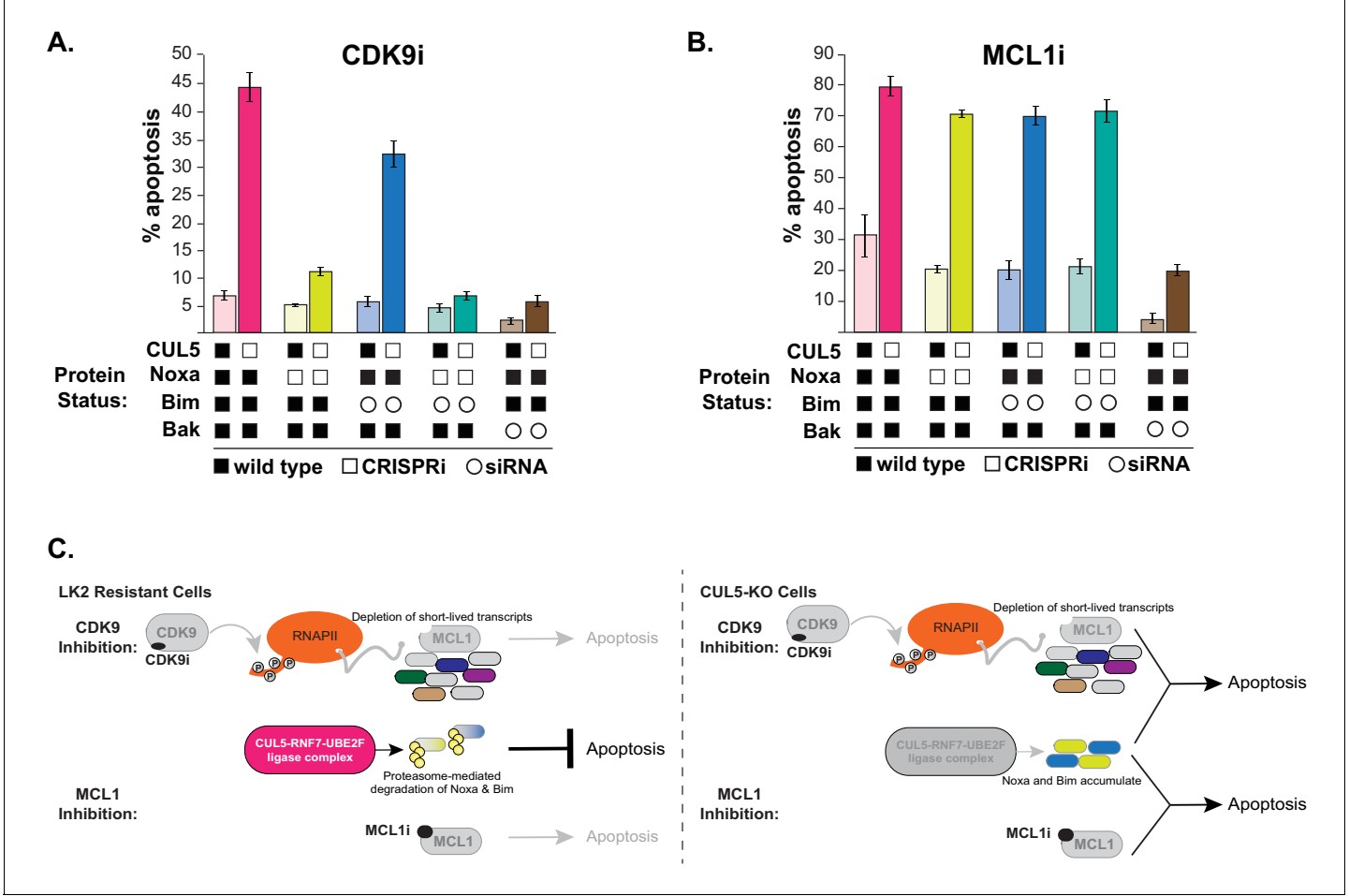

**Figure 5.** CUL5-depleted cells regain resistance to CDK9i, but not MCL1i, when Noxa is knocked down. (**A**) CUL5 and Noxa were knocked down in dCas9-KRAB-expressing LK2 cells by selecting for corresponding sgRNAs. Bim and Bak were knocked down by siRNA. Apoptosis was measured using Cell Event detection by flow cytometry after treatment with 3 µM CDK9i for 6 hr. Error bars are standard deviations of 3 biological drug treatment replicates. (**B**) Genetic manipulations were performed and apoptosis was measured as in (**A**) following MCL1i treatment (1 µM for 12 hr). (**C**) Schematic illustrating potential mechanisms of resensitization of resistant NSCLC lines to inhibition of MCL1.

DOI: https://doi.org/10.7554/eLife.44288.020

The following figure supplement is available for figure 5:

**Figure supplement 1.** CUL5-kd cells regain resistance to CDK9i when Noxa is depleted.

DOI: https://doi.org/10.7554/eLife.44288.021

CRL5 knockdown/out contributes to CDK9i sensitivity. However, more work is needed to elucidate the mechanistic distinctions by which CRL5 depletion re-sensitizes cells to both CDK9i and MCL1i (*Figure 5C*).

## Discussion

Cullin-RING complexes comprise the largest class of ubiquitin ligases and regulate a diverse array of biological processes (*Petroski and Deshaies, 2005*). While CUL3 is implicated in a wide array of cancer biologies, relatively little is known about the physiological functions of many CUL5-containing ubiquitin ligases. A few reports have implicated genetic alterations in CUL5 in cancer progression and our unbiased screens reveal that the CUL5-RNF7-UBE2F ubiquitin ligase can modulate the apoptotic threshold of LK2 cells in response to multiple emerging cancer therapeutics (*Burnatowska-Hledin et al., 2004*; *Fay et al., 2003*; *Lubbers et al., 2011*; *Xu et al., 2012*). The CUL5 ligase

complex regulates the stability of Noxa and Bim, while increased Noxa abundance through disruption of CUL5 underlies re-sensitization to a CDK9 inhibitor.

The discovery that inactivation of the CRL5 ligase can resensitize LK2 cells to both CDK9 and MCL1 inhibition suggests a possible co-treatment strategy, as well as a potential biomarker for patient stratification. Based on our data, patients with lung cancers that exhibit loss of function mutations in either CUL5, RNF7, UBE2F or Elongin B that coincide with high expression of MCL1/ Bcl-xL could be good candidates for treatment with CDK9i or MCL1i. These patients may be unlikely to have pre-existing CRL5-mediated resistance to treatment.

There are currently no small-molecule inhibitors that target specific components of the CRL5 complex. However, several potent inhibitors that broadly inhibit the ubiquitin-proteasome system have been developed. MLN4924 inhibits the NEDD8 activating enzyme (NAE), thereby preventing conjugation of NEDD8 onto all cullin scaffolds, and shows significant antitumor activity in clinical trials (*Soucy et al., 2009*). Similiarly, treatment with MLN7243, a small molecule inhibitor of the ubiquitin activating enzyme (UAE) currently in clinical trials, depletes cellular ubiquitin conjugates and also demonstrates antitumor activity in xenografts (*Hyer et al., 2018*). Several clinically approved proteasome inhibitors such as bortezomib and carfilzomib are currently used in the treatment of multiple myeloma and lymphomas (*Kuhn et al., 2007*; *Richardson et al., 2005*). Although a therapeutic window exists for these compounds, patients also experience adverse side effects due to the widespread effects of proteasome inhibition, complicating the possibility of co-dosing with CDK9i or MCL1i.

Compounds have also been developed that directly manipulate protein-protein interactions within specific ligases. The small molecule CC0651 selectively inhibits Cdc34A, an E2 for SCF-type CRLs, and has been shown to impede proliferation of cancer cells (*Ceccarelli et al., 2011*; *Huang et al., 2014b*). Another example is the synthesis of a potent inhibitor (ligand 7) targeting the von Hippel-Landau (VHL) substrate adaptor, thereby preventing the degradation of HIF-1α, a potential treatment for chronic anemia (*Galdeano et al., 2014*). While ubiquitin ligases are challenging targets for small molecule development, their importance in myriad disease states have spurred efforts to manipulate their activity. Our results in LK2 cells suggest that targeting CRL5 could be used to combat innate and acquired resistance to therapeutics that directly or indirectly affect MCL1. Additional work will be necessary to establish the mechanistic distinctions between CRL5's ability to set the apoptotic response to distinct small molecules, as well as the clinical applicability of targeting CRL5 in cancer.

## Materials and methods

### Cell culture and reagents

The LK2 cell line was obtained from AstraZeneca and maintained in RPMI supplemented with 10% (v/v) fetal bovine serum, 100 units/mL penicillin and streptomycin, GlutaMAX, sodium pyruvate and non-essential amino acids. All cells tested negative for Mycoplasma contamination using the MycoAlert Plus Mycoplasma detection kit from Lonza. AstraZeneca licensed the CDK9 inhibitor (AZD5576) originally published as PC585 (*Garcia-Cuellar et al., 2014*) and supplied it to us for this study. The MCL1 inhibitor (AZD5991) is synthesized by AstraZeneca (*Tron et al., 2018*).

### Isogenic CRISPR-Cas9 knockouts

Purified Cas9 ribonucleoproteins (RNPs) complexed with sgRNA pairs flanking coding exon one for CUL5 or UBE2F were nucleofected into LK2s to completely disrupt all coding potential as previously described (*Lingeman et al., 2017*). Nucleofected pools were seeded at single cell densities by FACS sorting. Clones were initially screened by western blot for loss of protein and confirmed by Sanger sequencing.

Individual analysis of sgRNA phenotypes sgRNA protospacers targeting CUL5, RNF7, UBE2F, Elongin B, Elongin C, Bcl-xL, Noxa and a negative control non-targeting protospacer were individually cloned into BstXI/BlpI-digested pCRISPRia-v2 (Addgene #8432) by ligating annealed complementary synthetic oligonucleotide pairs. The sgRNA expression vectors were packaged into lentivirus as previously described and successful transductants were selected with puromycin at a final concentration of 2.5 μg/mL. Protospacer sequences are in *Supplementary file 1*.

## Lentivirus production and transduction

Lentivirus was produced by transfecting HEK293T with standard packaging vectors using the TransIT-LT1 Transfection Reagent (MIR 2306; Mirus Bio LLC). Viral supernatant was collected at 48 hr and 72 hr after transfection, filtered through a 0.45 µm polyethersulfone syringe filter, snap-frozen and stored at −80℃ for future use. For screens, viral titrations were performed by transducing LK2 cells at serial dilutions and assessing BFP (present on the sgRNA-encoding plasmid) percentages 48 hr following transduction. The viral dilution resulting in ~15% BFP-positive cells was used to transduce cells for the screens.

## Genome-scale CRISPRi screens

The human CRISPRi v2 library contains 5sgRNAs/annotated TSS of each gene comprising a total of 104,535 sgRNAs. In order to maintain 500X coverage (representation) of each guide at all times, ~$50 \times 10^6$ cells need to be transduced with one sgRNA. To ensure delivery of one sgRNA/cell a low MOI of ~0.3 is used, which results in not every cell getting transduced, thus 5–6 fold more cells need to be transduced in order to maintain 500X coverage. Replicate cultures of $3 \times 10^8$ cells were plated in twenty 15 cm dishes and transduced at an MOI of ~0.3 in the presence of 4 µg/mL polybrene. Cells were split into 2.5 µg/mL puromycin and selected for 4 days. Cells were passaged into regular media and recovered for 2 days. $60 \times 10^6$ were collected as the 'background' sample. $60 \times 10^6$ cells were treated with 3 µM CDK9i and 0.5 µM Cell Event for 6 hr. $60 \times 10^6$ cells were treated with 1 µM MCL1i and 0.5 µM Cell Event for 12 hr. $60 \times 10^6$ cells were treated with DMSO (1:10,000X dilution) and 0.5 µM Cell Event for 12 hr. After treatment, cells were trypsinized, washed and fixed in 1% formaldehyde/PBS. The entirety of the cell population was then FACS-sorted where GFP-positive (apoptosing) cells were isolated. As previously mentioned, this was done in duplicate. Genomic DNA was purified from each cell population with blood purification kits (Machery-Nagel, Nucleospin blood L or XL, depending on cell number) and the sgRNA-encoding region was enriched, amplified, and processed for sequencing on the Illumina Hiseq 2500. TruSeq index sequences unique to each cell population were used to multiplex samples.

## Pooled screen analysis

Data analysis was performed as described (*Horlbeck et al., 2016*; *Li et al., 2014*). Briefly, sequence reads from the Illumina HiSeq 2500 were trimmed, aligned to CRISPRi v2 sgRNA library, counted and normalized. For the Python-based ScreenProcessing pipeline, sgRNA phenotypes and negative control gene phenotypes were determined along with Mann-Whitney P values. The top three guide-level phenotypes were collapsed to produce the gene-level phenotype score. For genes with multiple annotated transcription start sites (TSSs), phenotypes were calculated for each TSS and the TSS with the lowest Mann Whitney p-value was used to represent the gene. For MAGeCK, all sgRNAs/gene (including both TSSs if there are two) are ranked based on p-values from mean variance modeling. A robust ranking aggregation (RRA) algorithm is then used to call genes that are significantly enriched or depleted based on p-value and false discovery rate (FDR).

## qRT-PCR

For qPCR, RNA was extracted with the RNeasy Mini Kits (Qiagen). cDNA was produced from 1 µg of purified RNA using the iScript Reverse Transcription Supermix for RT-qPCR (Bio-Rad Laboratories). qPCR reactions were performed with the SsoAdvanced Universal SYBR Green Supermix (Bio-Rad Laboratories) in a total volume of 10 µl with primers at final concentrations of 500 nM. Primer sequences are included in *Supplementary file 2*. The thermocycler was set for 1 cycle of 95℃ for 30 s, and 40 cycles of 95℃ for 5 s and 55℃ for 15 s, respectively. Fold enrichment of the assayed genes over the control *HPRT* and/or *GAPDH* loci were calculated using the $2^{-\Delta\Delta C}T$ method as previously described (*Livak and Schmittgen, 2001*).

## Cell event apoptosis detection assay

Cell Event reagent was added to cells for the duration of the drug or vehicle treatment, at a final concentration of 0.5 µM. Cells were harvested by trypsinization and analyzed on a flow cytometer, either the BD FACSARIA for the screen or an Attune Nxt cytometer (ThermoFisher) for the follow-up experiments.

## Caspase activation assay

Cells were plated at 5000 cells/well of a 384-well white opaque plates in corresponding cell growth media. Cells were treated with compounds at indicated concentrations for 6 hr (37°C, 5% $CO_2$) with a final DMSO concentration of 0.3%. Caspase-3/7 activation was subsequently determined using a Caspase-Glo 3/7 Reagent (Promega Corporation) as described in manufacturer's instructions. Dose-response curves were plotted and analyzed (including EC50 determination) using GraphPad Prism. Percentage of caspase activation was calculated against the maximum caspase activation value (100%) obtained with a combination of MCL1, BCL2, and Bcl-xL inhibition.

## Cell viability assay

Cells were plated at 5,000 cells/well of a 384-well white opaque plates in corresponding cell growth media. Cells were treated with compounds at indicated concentrations for 24 hr (37°C, 5% $CO_2$) with a final DMSO concentration of 0.3% and assayed for viability using the CellTiter-Glo Reagent (Promega Corporation) as described in manufacturer's instructions. Results were normalized to the samples without treatment at time 0. $GI_{50}$ values were calculated using nonlinear regression algorithms in Prism.

## Immunoblotting

Primary antibodies against the following proteins were used: CUL5 (ab184177; Abcam); RNF7 (ab181986; Abcam); UBE2F (ab185234; Abcam); Noxa (OP180; Millipore); Bim (ab32158; Abcam); Bid (ab32060; Abcam); Puma (ab9643; Abcam); Bad (ab62465; Abcam); Bik (ab52182; Abcam); Beclin (ab207612; Abcam); Bmf (ab9655; Abcam); MCL1 (D35A5 #5453; CST); Bcl-xL (54H6; CST); c-PARP (19F4 #9546; CST) Ubiquitin (P4D1 #3936; CST); HA (C29F4; CST); GAPDH (14C10 #2118; CST); p53 (DO-1 sc-126; Santa Cruz Biotechnology); HSP90 (sc-69703; Santa Cruz Biotechnology). For each protein antibody, manufacturer's recommended dilutions were used. Mouse or rabbit immunoglobulin G was visualized at a 1:10,000 dilution: donkey anti-mouse 680 (925–68022; LI-COR); donkey anti-rabbit 680 (925–68023; LI-COR); donkey anti-mouse 800 (925–32212; LI-COR); donkey anti-rabbit 800 (925–32213; LI-COR). Blots were imaged on an Odyssey CLx Imaging System (LI-COR).

## Immunoprecipitation

LK2 wild type and CUL5-KO cells were transfected with equal amounts of His-Ubiquitin and either HA-Noxa or HA-Bim. 48 hr after transfection, cells were treated with 100 µM epoxomicin for 8 hr. Cells were harvested and lysed in either denaturing buffer (8M Urea, 300 mM NaCl, 0.5% NP-40, 50 mM Na2HPO4, 50 mM Tris pH8.0) or lysis buffer from ThermoFisher (Pierce HA-Tag Magnetic IP/ CoIP Kit #88838). Denatured extracts were bound to magnetic Ni-NTA beads (Qiagen #36111), while other lysed extracts were bound to magnetic anti-HA beads. Beads were washed five times and eluted in 1X Laemmli buffer by boiling for 5 min. Samples were loaded on a gel and processed for immunoblotting.

## siRNA treatment

Cells were reverse-transfected in six-well plates using RNAiMAX (Thermo Fisher Scientific) with 50 nM siRNA. 48 hr following siRNA treatment, cells were treated for the Cell Event apoptosis assay as indicated and also harvested to verify knockdown by qRT-PCR. BCL2L11 (Bim) siRNA SMARTpool was from Dharmacon (L-004383-00-0005) and BAK siRNA was obtained from Ambion (Life Technologies 4457298).

## Clonogenic assay

Six-well plates were seeded at 500 cells/well and allowed to attach for 12 hr. Cells were treated with DMSO or 1 µM CDK9i or 1 µM MCL1i for 8 hr. Drug treatments were washed out and plates were replenished with fresh media. 2 weeks later colonies were stained with 0.5% crystal violet and 6% glutaraldehyde in water. Plates were scanned using a flatbed scanner and colonies were scored on ImageJ.

## Acknowledgements

This work was supported by a National Institutes of Health New Innovator Awards (DP2 HL141006), the Li Ka Shing Foundation, the Heritage Medical Research Institute, the California Institute of Regenerative Medicine (DISC1-08776), and funded research support from AstraZeneca. This work used the Vincent J Coates Genomics Sequencing Laboratory at UC Berkeley, supported by the NIH S10 OD018174 Instrumentation Grant.

## Additional information

### Competing interests

Justin Cidado, Courtney Andersen, Cortni Dick, Matthew A Belmonte, Lisa Drew: employed by AstraZeneca, from whom funded research support was received. The other authors declare that no competing interests exist.

### Funding

| Funder | Grant reference number | Author |
| --- | --- | --- |
| AstraZeneca | | Shaheen Kabir<br>Justin Cidado<br>Courtney Andersen<br>Cortni Dick<br>Pei-Chun Lin<br>Hong Ma<br>Matthew A Belmonte<br>Lisa Drew<br>Jacob E Corn |
| National Institutes of Health | DP2 HL141006 | Jacob E Corn |
| Li Ka Shing Foundation | | Jacob E Corn |
| Heritage Medical Research Institute | | Jacob E Corn |
| California Institute for Regenerative Medicine | DISC1-08776 | Shaheen Kabir<br>Seung Hyun Baik<br>Jacob E Corn |

The funders had no role in study design, data collection and interpretation, or the decision to submit the work for publication.

### Author contributions

Shaheen Kabir, Conceptualization, Formal analysis, Validation, Investigation, Visualization, Methodology, Writing—original draft, Project administration, Writing—review and editing; Justin Cidado, Conceptualization, Investigation, Methodology, Project administration, Writing—review and editing; Courtney Andersen, Investigation, Methodology; Cortni Dick, Hong Ma, Seung Hyun Baik, Validation, Investigation; Pei-Chun Lin, Conceptualization, Investigation; Therese Mitros, Data curation, Software, Visualization; Matthew A Belmonte, Conceptualization, Investigation, Methodology; Lisa Drew, Conceptualization, Project administration; Jacob E Corn, Conceptualization, Supervision, Funding acquisition, Writing—review and editing

### Author ORCIDs

Shaheen Kabir https://orcid.org/0000-0002-9035-1124
Courtney Andersen http://orcid.org/0000-0003-2064-2273
Jacob E Corn https://orcid.org/0000-0002-7798-5309

### Decision letter and Author response

Decision letter https://doi.org/10.7554/eLife.44288.028
Author response https://doi.org/10.7554/eLife.44288.029

## Additional files

### Supplementary files

• Supplementary file 1. Protospacer sequences.
DOI: https://doi.org/10.7554/eLife.44288.022

• Supplementary file 2. Primer sequences.
DOI: https://doi.org/10.7554/eLife.44288.023

• Transparent reporting form
DOI: https://doi.org/10.7554/eLife.44288.024

### Data availability

Sequencing data are available on NCBI BioProject under accession number PRJNA553254.

The following dataset was generated:

| Author(s) | Year | Dataset title | Dataset URL | Database and Identifier |
|---|---|---|---|---|
| Kabir S | 2019 | Genome-wide CRISPRi Resensitization Screens with MCL1 Inhibitors | https://www.ncbi.nlm.nih.gov/bioproject/?term=PRJNA553254 | NCBI BioProject, PRJNA553254 |

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
