## [Decision Letter]

Thank you for submitting your article "The CUL5 ubiquitin ligase complex mediates resistance to CDK9 and MCL1 inhibitors in lung cancer cells" for consideration by *eLife*. Your article has been reviewed by Ivan Dikic as the Senior Editor, Michael Green as the Reviewing Editor, and two reviewers. The following individual involved in review of your submission has agreed to reveal their identity: Narendra Wajapeyee (Reviewer #1).

The reviewers have discussed the reviews with one another and the Reviewing Editor has drafted this decision to help you prepare a revised submission.

Summary:

In this manuscript, Kabir et al., address the important problem of resistance/non-response to drugs under clinical development. In a CRISPRi/FACS screen they identify genes for which knock-down synergizes with either CDK9 (CDK9i) or MCL1 (MCL1i) inhibition. Along BclXl, which is known to synergize with MCL1 inhibition and lower the apoptotic threshold, they identify components of a Cullin5 RING ubiquitin ligase (CRL5), namely the scaffold protein Cul5, the zinc-finger protein RBX2/RNF7 and the NEDD8 specific E2 enzyme UBE2F, suggesting a role in a CRL5 type ligase in regulating apoptosis. They go on to validate these findings in cellular models using individual knock-outs/downs of either CUL5, UBE2F and RNF7 confirming the role of these proteins in regulating the sensitivity to CDK9i/MCL1i. Since substrate receptors are a key component of a CRL5 ligase and confer specificity to the ligase complex, they attempt to identify the relevant substrate receptor by targeted knock-out of all known CRL5 substrate receptors, however, fail to identify a specific substrate receptor. In search of the relevant downstream targets, they observe up-regulation of Bim and Noxa in response to CRL5 inhibition by either CRISPRi or shRNA. They further show that in a CUL5 depleted background, efficacy of CDK9i but not MCL1i depends on the presence of Noxa.

In summary, the authors identify CRL5 as a critical node in regulating apoptosis and suggest that targeting CRL5 could provide potential synergy with CDK9i/MCL1i, or that patient stratification based on mutations in components of the CRL5 ligase could inform benefits of CDK9i/MCL1i. In the absence of a known substrate receptor, the authors did not succeed in identifying the mechanism by which CRL5 depletion regulates apoptosis, and the differences in response of CDK9i vs MCL1i suggest a more complex regulation of the pathway.

Essential revisions:

1) The manuscript is based on the assumption that a key target of CDK9i is MCL1 (Abstract: "that interrogate two drugs directly or indirectly targeting MCL1"), which appears to be only marginally supported. If that were to be the case, the genome scale screens should exhibit a significant global correlation, which the authors should show. It is also confusing why in such a scenario CDK9i but not MCL1i requires Noxa (Discussion section). The authors need to be clearer about their assumptions and the nature of CDK9i leading to a global and profound downregulation of transcription with likely complex downstream effects directly/or indirectly signaling into apoptosis. The authors need to discuss this more accurately, and reference data that suggests that CDK9i exerts therapeutic effects pre-dominantly through MCL1 downregulation.

2) While the manuscript addresses an important problem and the screens are well designed and rigorously executed, it appears that some of the downstream experiments (in particular the investigation of Bim and Noxa) are not sufficient to support the overall statements. For example, subsection "Noxa is required to resensitize CUL5-deficient cells to MCL1 inhibition", however, in Figure 5B the authors show that co-depletion of CUL5 and Noxa still results in apoptosis levels comparable to CUL5-/- upon MCL1i treatment. There are more such inconsistencies that need to be addressed.

3) Overall, the authors should rewrite the manuscript and be clearer about their overall conclusions. They show that depletion of key components of the CRL5 family of ubiquitin ligases can sensitize to CDK9 and MCL1 inhibition, while more work is needed to identify the exact mechanism.

4) The majority of the drug-resistance experiments are short-term apoptosis assays. To draw strong conclusion about drug-resistance and susceptibility it is important to perform some type of long-term survival and proliferation assays. Please confirm the key conclusions of the study using some type of long-term survival and proliferation assay.

[Editors' note: further revisions were requested prior to acceptance, as described below.]

Thank you for resubmitting your work entitled "The CUL5 ubiquitin ligase complex mediates resistance to CDK9 and MCL1 inhibitors in lung cancer cells" for further consideration at *eLife*. Your revised article has been favorably evaluated by Cynthia Wolberger (Senior Editor), a Reviewing Editor, and two reviewers.

The manuscript has been improved but there are some remaining issues that need to be addressed before acceptance, as outlined below:

In particular, one of the reviewers feels that the conclusions, particularly the clinical implications, have been over-stated. Some of the reviewer's original comments and other comments provided during discussion of the manuscript are provided below.

Please carefully review these comments and revise your manuscript accordingly. Please provide a point-by-point response of how you have revised the manuscript to address the reviewer's concerns.

Comments from original review:

In this revised manuscript, Kabir et al., deploy elegant CRISPR screens to identify potential mechanisms of resistance/non-response to CDK9 inhibitors (CDK9i) and MCL1 inhibitors (MCL1i). They identify components of the Cullin-5 RING ubiquitin ligase complex (CRL5) as mediators of sensitivity to CDK9i and MCL1i. They further show that genetic inactivation of CUL5 or RNF7 leads to elevated levels of the pro-apoptotic proteins BIM and NOXA. While they show that knockout of NOXA partially abrogates the sensitizing effect of CUL5 knockout in the context of CDK9i, no effect is observed for MCL1i.

The key criticism to the manuscript was the weakly supported assumption that both, CDK9i and MCL1i, act predominantly through inhibition of MCL1. The authors acknowledge the lack of data for their initial assumption yet fail to provide data that would further support this key assumption for conclusions drawn in the paper (rebuttal major points 1-3). In light of this, I feel that both screens have to be considered as potentially correlated but not necessarily functionally linked and the key message that CUL5 ubiquitin ligase complexes mediate resistance to CDK9i and MCL1i is not sufficiently supported. With just a single experiment in a single cell line, and sensitization to potentially unrelated drugs, depletion of CUL5 could result in general lowering of the apoptotic threshold instead of a CDK9i/MCL1i specific effect.

Overall, I feel this manuscript as it is presented right now is overselling the therapeutic value of the findings and more thorough experiments and importantly discussion of the data would be necessary.

Representative major points:

The screens are carried out in a single cell line, and since at least for MCL1i no mechanistic understanding beyond the genetic link to CUL5 is provided (regulation of NOXA impacts CDK9i but not MCL1i), the obvious question is whether knockdown of CUL5, RNF7 or UBE2F is just a general reduction of apoptotic threshold and as such would sensitize without necessarily increasing the therapeutic index. Moreover, I feel the overlap between both screens is overstated at multiple occasions, for example in Figure 2—figure supplement 1D,E referenced as "the top resensitization hits were strikingly consistent, only 3 out of the top 10 hits overlapped.

For MCL1i, the authors provide only a genetic screen, which while elegant, does not provide mechanistic inside into the resistance mechanism. The fact that Noxa/Bim are downregulated appears to be correlated, but not necessarily functionally linked to MCL1i.

In the second paragraph of the Discussion section – refer to significant co-occurrence of MCL1 amplification with CUL5/UBE2F deletions, which appears to be overstated based on data available at cbioportal: Out of 3351 lung cancer samples, only 3 have a co-occurrence of CUL5 deletions with either MCL1 or BCL^-^xL amplification, 2 a co-occurrence of UBE2F deletion with MCL1/BCL^-^xL amplification, and none a co-occurrence of RNF7 deletions. If further data is available, it should be provided as supplementary figure.

---

## [Author Response]

Essential revisions:1) The manuscript is based on the assumption that a key target of CDK9i is MCL1 (Abstract: "that interrogate two drugs directly or indirectly targeting MCL1"), which appears to be only marginally supported. If that were to be the case, the genome scale screens should exhibit a significant global correlation, which the authors should show.

We agree with the reviewers that while MCL1 is a key target of acute CDK9 inhibition, there are numerous additional targets that are concomitantly depleted by transcriptional inhibition. We have stated this more clearly in the Introduction and have also clarified that our goal is to find any synergistic hits that can be used to stratify patients or in combination therapies with either inhibitor. Furthermore, we have also included a Venn diagram showing that while a subset of hits in the MCL1i screen overlap with the CDK9i screen, the CDK9i screen has a greater number of hits likely due to its additional targets (Figure 2—figure supplement 1C, discussed in subsection “Genome-wide CRISPRi screens identify factors that synergize with CDK9 and MCL1 inhibition”).

It is also confusing why in such a scenario CDK9i but not MCL1i requires Noxa (Discussion section). The authors need to be clearer about their assumptions and the nature of CDK9i leading to a global and profound downregulation of transcription with likely complex downstream effects directly/or indirectly signaling into apoptosis. The authors need to discuss this more accurately, and reference data that suggests that CDK9i exerts therapeutic effects pre-dominantly through MCL1 downregulation.

We also agree with the reviewers that the dependency on Noxa for eliciting apoptosis in CDK9i treated cells, but not MCL1i-treated cells is likely due to the polypharmacology (e.g. widespread transcriptional inhibition) of CDK9i. We have amended the text to stress this point and hopefully clarify for the readers the reason for this discrepancy.

2) While the manuscript addresses an important problem and the screens are well designed and rigorously executed, it appears that some of the downstream experiments (in particular the investigation of Bim and Noxa) are not sufficient to support the overall statements. For example, subsection "Noxa is required to resensitize CUL5-deficient cells to MCL1 inhibition", however, in Figure 5B the authors show that co-depletion of CUL5 and Noxa still results in apoptosis levels comparable to CUL5-/- upon MCL1i treatment. There are more such inconsistencies that need to be addressed.

We apologize for the confusion and our imprecise phrasing in interchangeably using MCL1 inhibition to describe treatment with CDK9 inhibitor that indirectly reduces levels of MCL1. We have attempted to make these statements much clearer (Introduction, Results section, Discussion section, Figure 3 legend and Figure 3—figure supplement 2 legend).

3) Overall, the authors should rewrite the manuscript and be clearer about their overall conclusions. They show that depletion of key components of the CRL5 family of ubiquitin ligases can sensitize to CDK9 and MCL1 inhibition, while more work is needed to identify the exact mechanism.

We have rewritten the manuscript to make it more concise and focused on clarifying the important points the reviewers have raised. We have also rephrased our conclusions as follows and hope that the reviewers find this to be clearer.

“In conclusion, we find that depletion of key components of the CRL5 family of ubiquitin ligases sensitizes otherwise resistant cells to CDK9 and MCL1 inhibition. We show that the CRL5 complex targets pro-apoptotic proteins Noxa and Bim for degradation, which may contribute in part to the resistance against CDK9i and MCL1i, but more work is needed to elucidate the exact mechanistic distinctions between direct and indirect inhibition of MCL1.”

4) The majority of the drug-resistance experiments are short-term apoptosis assays. To draw strong conclusion about drug-resistance and susceptibility it is important to perform some type of long-term survival and proliferation assays. Please confirm the key conclusions of the study using some type of long-term survival and proliferation assay.

This is an excellent point that we have now addressed through new clonogenic experiments presented in the revised Figure 3G-H. Clonogenic assays are established in the field of cancer biology as readouts of long-term survival, replicative capacity and determining the effects of anti-cancer therapeutics on colony forming ability. We performed clonogenic assays using LK2 parental cells and our isolated CUL5-KO and UBE2F-KO clones. Cells were subjected to acute CDK9i and MCL1i treatments, since cells do not tolerate persistent treatment as discussed in the manuscript. The results clearly show that LK2 cells are much more resistant to these small molecules than the CUL5-KO and UBE2F-KO cells. We thank the reviewers for this suggestion and feel that the results of our clonogenic assays strengthen the manuscript, providing credence to our conclusion that CRL5 does in fact sensitize cells in a permanent way to CDK9 and MCL1 inhibition.

[Editors' note: further revisions were requested prior to acceptance, as described below.]

Comments from original review:In this revised manuscript, Kabir et al., deploy elegant CRISPR screens to identify potential mechanisms of resistance / non-response to CDK9 inhibitors (CDK9i) and MCL1 inhibitors (MCL1i). They identify components of the Cullin-5 RING ubiquitin ligase complex (CRL5) as mediators of sensitivity to CDK9i and MCL1i. They further show that genetic inactivation of CUL5 or RNF7 leads to elevated levels of the pro-apoptotic proteins BIM and NOXA. While they show that knockout of NOXA partially abrogates the sensitizing effect of CUL5 knockout in the context of CDK9i, no effect is observed for MCL1i.

We thank the referees for critically re-reading our manuscript and for their support in the revision process. Most of the referee’s comments relate to possibly overstating the importance or functional relevance of the CRL5 complex in the context of CDK9i and MCL1i. We certainly did not mean to “oversell” the data, and most of our edits to the first revision were aimed at softening our interpretation. We have further scaled this back throughout the manuscript, and hope that the renewed focus on the data itself has now addressed the referee’s concern. In one case we have chosen to add a statistical measure to strengthen the data presented (e.g. overlap between screen outputs). We note that a new preprint (June 23, 2019) from the lab of John Doench has recently used CRISPR screening to identify CUL5, RNF7, and UBE2F as synthetic lethal with MCL1 knockout (notably these genes were *not* synthetic lethal hits in screens with several other parental knockouts). This preprint (which cites our own preprint), highlights the reproducibility of our results, its specificity to MCL1 intervention, and stresses the timeliness of our manuscript (https://www.biorxiv.org/content/biorxiv/early/2019/06/23/677385.full.pdf).

The key criticism to the manuscript was the weakly supported assumption that both, CDK9i and MCL1i, act predominantly through inhibition of MCL1. The authors acknowledge the lack of data for their initial assumption yet fail to provide data that would further support this key assumption for conclusions drawn in the paper (rebuttal major points 1-3).

We apologize for inadvertently suggesting that CDK9i primarily acts through inhibition of MCL1. This was not what we meant, and we thank the referee for pointing this out. We instead mean that one of CDK9i’s mechanisms is through depletion of MCL1 transcript. In this way, CDK9i is an indirect inhibitor of MCL1 that also downregulates other transcripts relevant to cancer (e.g. *MYC*), while MCLi is a direct inhibitor of MCL1. Figure 4D and Figure 3—figure supplement 2E show that CDK9i in our hands potently depletes MCL1 mRNA and protein. We did not provide further data on the ability of CDK9 inhibitors to downregulate *MCL1* and thereby alter tumor cell survival due to the extensive literature on this topic (e.g. Brisard et al., 2018, Minzel et al., 2018, Inoue-Yamauchi et al., 2017, Tong et al., 2017, Mitra et al., 2016, Baker et al., 2016, Gregory et al., 2015, Yin et al., 2014, Lemke et al., 2014, Thomas et al., 2013, and MacCallum et al., 2005). We have added language attempting to clarify that CDK9i effects MCL1 but also other oncogenes (see Introduction).

In light of this, I feel that both screens have to be considered as potentially correlated but not necessarily functionally linked and the key message that CUL5 ubiquitin ligase complexes mediate resistance to CDK9i and MCL1i is not sufficiently supported. With just a single experiment in a single cell line, and sensitization to potentially unrelated drugs, depletion of CUL5 could result in general lowering of the apoptotic threshold instead of a CDK9i/MCL1i specific effect.Overall, I feel this manuscript as it is presented right now is overselling the therapeutic value of the findings and more thorough experiments and importantly discussion of the data would be necessary.

We agree with the referee that depletion of CUL5 could generally reduce the apoptotic threshold. Indeed, anti-apoptotic BCL^-^2 family members such as MCL1 are responsible for setting the apoptotic threshold, so our data are not at odds with the referees’ comments. Inhibition or depletion of BCL^-^2, BCL^-^xL, and MCL1 synergizes with a variety of anti-cancer agents (inhibitors of kinases such as BRAF, BTK, etc and even HDAC inhibitors). Inhibition of BCL2 family members such as MCL1 is particularly important for primary effect in contexts where the family member is highly upregulated. Our data suggest that depletion of the CRL5 complex can overcome the extremely high apoptotic threshold set by simultaneous BCL^-^xL overexpression and MCL1 copy number gain (Figure 1B, Supplementary file 1). Depletion of CRL5 does not induce high basal levels of apoptosis, as is typical of BCL^-^xL inhibition, but instead strongly synergizes with both MCL1i and CDK9i (Figure 3E). Finally, we note that a very recent preprint from the lab of John Doench describes synthetic lethal screens for genetic interactions with multiple factors. When querying MCL1 (and *only* MCL1), the authors found CUL5, UBE2F, and RNF7 as top synthetic lethal hits. This genetic interaction underscores our own results with chemical inhibitors. DeWeirdt et al., 2019.

Inhibition of BCL2 family members has proven to have therapeutic value, despite associated toxicity in certain settings. But proving this therapeutic value took many years and many studies, and we fully acknowledge that our current manuscript is not the final word on whether targeting the CRL5 complex is therapeutically appropriate. We have edited the manuscript to attempt to adequately represent the referee’s point and to avoid overselling our current results, while still attempting to convey their importance. For example, we have stressed that our data demonstrate that CRL5 knockdown re-sensitizes to both CDK9i and MCL1i in one NSCLC setting, but other mechanisms could underlie resistance of other NSCLC lines.

Representative major points:The screens are carried out in a single cell line, and since at least for MCL1i no mechanistic understanding beyond the genetic link to CUL5 is provided (regulation of NOXA impacts CDK9i but not MCL1i), the obvious question is whether knockdown of CUL5, RNF7 or UBE2F is just a general reduction of apoptotic threshold and as such would sensitize without necessarily increasing the therapeutic index.

See comments above about apoptotic threshold. Particularly, note that depletion of CRL5 does not induce high basal levels of apoptosis, as is typical of BCL^-^xL inhibition, but instead strongly synergizes with both MCL1i and CDK9i (Figure 3E). Since BCL^-^xL inhibitors are already used therapeutically (but with significant toxicity), we feel that this implies that targeting CRL5 could be useful in future. We also note that a very recent preprint from the lab of John Doench describes synthetic lethal screens for genetic interactions with multiple factors. When querying MCL1 (and *only* MCL1), the authors found CUL5, UBE2F, and RNF7 as top synthetic lethal hits. This genetic interaction underscores our own results with chemical inhibitors. DeWeirdt et al., 2019.

We fully acknowledge that our current manuscript is not the final word on whether targeting the CRL5 complex is therapeutically appropriate. This will take many years and many studies. We have attempted to clarify this point in the Discussion section.

Moreover, I feel the overlap between both screens is overstated at multiple occasions, for example in Figure 2—figure supplement 1D,E referenced as "the top resensitization hits were strikingly consistent, only 3 out of the top 10 hits overlapped.

Given the genome-scale of the library, the overlap between hits in the CDK9i and MCL1i screens is highly significant. This can be calculated as a hypergeometric probability, conveniently implemented at the following website: http://nemates.org/MA/progs/overlap_stats.html.

Taking the cutoffs shown in Figure 2—figure supplement 1C, 18,746 genes in the CRISPRiv2 library, 24 hits with MCL1i, 117 hits with CDK9i, and 6 gene overlaps, the representation factor is 40.1 with p < 6.4e-9. As expected, CDK9i’s pleiotropic effects on transcription lead to more hits than the more specific MCL1i, but the overlap is significant. We furthermore note that Figure 2 —figure supplement 1D/E show the raw output of MaGECK, which ranks gene-level hits solely by statistical significance, neglects effect size, and only shows the ten lowest p-values (RRA is a type of p-value). Of course, relevant hits can be present with p-values larger than the lowest observed p-values while still passing an appropriate threshold. This is why we presented volcano plots for both screens in Figure 2, taking into account both statistical significance and effect size.

We have added information on the statistical significance of the overlap to the revised manuscript. We have also revised the statements about the screens in an attempt to avoid over-stating the implications of our data.

For MCL1i, the authors provide only a genetic screen, which while elegant, does not provide mechanistic inside into the resistance mechanism. The fact that Noxa/Bim are downregulated appears to be correlated, but not necessarily functionally linked to MCL1i.

The reviewer is completely correct that our mechanistic insights for CRL5 downregulation in the context of CDK9i are stronger than CRL5 downregulation in the context of MCL1i. We have made edits throughout the manuscript to clarify this point.

In the second paragraph of the Discussion section refer to significant co-occurrence of MCL1 amplification with CUL5/UBE2F deletions, which appears to be overstated based on data available at cbioportal: Out of 3351 lung cancer samples, only 3 have a co-occurrence of CUL5 deletions with either MCL1 or BCL^-^xL amplification, 2 a co-occurrence of UBE2F deletion with MCL1/BCL^-^xL amplification, and none a co-occurrence of RNF7 deletions. If further data is available, it should be provided as supplementary figure.

This discussion point was originally based on TCGA lung adenocarcinomas published in Nature in 2014 (doi: 10.1038/nature13385), where out of 230 samples profiled, 5 samples had a co-occurrence of MCL1 amplification and CUL5 mutation/deletion. However, the reviewer makes a fair point that as we do not observe a more significant trend when taking into account all the 3351 lung cancer samples available, this discussion point is overstated. We thank the reviewer for carefully and critically assessing our manuscript and have removed this statement so as not to mislead.